# Impact of a strong volcanic eruption on the summer middle atmosphere in UA-ICON simulations

Sandra Wallis[1], Hauke Schmidt[2], and Christian von Savigny[1]

[1]Institute of Physics, University of Greifswald, Felix-Hausdorff-Str. 6, 17489 Greifswald, Germany
[2]Max Planck Institute for Meteorology, Bundesstrasse 53, 20146, Hamburg, Germany

**Correspondence:** Sandra Wallis (sandra.wallis@uni-greifswald.de)

**Abstract.** Explosive volcanic eruptions emitting large amounts of sulfur can alter the temperature of the lower stratosphere and change the circulation of the middle atmosphere. The dynamical response of the stratosphere to strong volcanic eruptions has been the subject of numerous studies. The impact of volcanic eruptions on the mesosphere is less well understood because of a lack of large eruptions in the satellite era and only sparse observations before that period. Nevertheless, some measurements indicated an increase in mesospheric mid-latitude temperatures after the 1991 Pinatubo eruption. The aim of this study is to uncover potential dynamical mechanisms that may lead to such a mesospheric temperature response. We use the upper-atmospheric icosahedral non-hydrostatic (UA-ICON) model to simulate the atmospheric response to an idealized strong volcanic injection of 20 Tg S into the stratosphere (about twice as much as the eminent 1991 Pinatubo eruption). Two experiments with differently parameterized effects of sub-grid scale orography are compared to test the impact of different atmospheric background states. The simulations show a significant warming of the polar summer mesospause of up to 15-21 K in the first November after the eruption. We argue that this is mainly due to intrahemispheric dynamical coupling in the summer hemisphere and potentially enhanced by interhemispheric coupling with the winter stratosphere. This study focuses on the first austral summer after the eruption, because mesospheric temperature anomalies are especially relevant for the properties of noctilucent clouds, whose season peaks around January in the southern hemisphere.

## 1 Introduction

Strong tropical volcanic eruptions that emit sulfur into the lower stratosphere can have a significant impact on the state of the atmosphere and Earth's climate (Marshall et al., 2022; Robock, 2000; Timmreck, 2012; Cole-Dai, 2010). $SO_2$ is oxidized to sulfuric acid, which easily forms sulfate particles by heteromolecular nucleation. These particles can scatter part of the incoming shortwave radiation back to space and lead to a reduction of the global surface temperature. Sulfate aerosol additionally absorbs longwave radiation resulting in a warming of the lower tropical stratosphere. These temperature anomalies have the potential to alter the middle atmospheric circulation. There is scientific consensus that some features of the stratospheric circulation are altered after large tropical eruptions, which is, however, largely built on modeling results because of the limited number of large eruptions during the satellite era. The Brewer-Dobson circulation (or meridional overturning circulation) is assumed to accelerate due to the tropical heating (e.g. Pitari et al. (2016)) and the westerlies of the polar vortices are assumed to

enhance in the first post-eruption boreal winter (e.g. Azoulay et al. (2021)). It is in general assumed that the cause of the vortex strengthening is the change of the meridional temperature gradient by the aerosol heating being strongest at low latitudes. Bittner et al. (2016), however, argue, that further wave-mean flow interaction must be involved in accelerating the vortex. Such an altered circulation of the stratosphere might influence the troposphere enhancing the probability of a so-called winter warming in northern Eurasia and North America, a feature that has been observed after several large eruptions (Robock and Mao, 1992).

Aerosol effects on stratospheric ozone may contribute to circulation changes (Stenchikov et al., 2002). The eruption-induced temperature perturbation of the atmosphere may also depend on the phase of the quasi-biennial oscillation (QBO) and the El Nino/Southern Oscillation phenomenon (Thomas et al., 2009).

The impact of volcanic eruptions on the mesosphere (approximately from 50-90/100 km altitude (Poppoff and Whitten, 1976; von Zahn et al., 1996; She and von Zahn, 1998)) is less well observed and investigated (von Savigny et al., 2020b).

There are some ground-based and satellite-born observations that indicate a warming of the mesosphere and mesopause region after the 1991 Pinatubo eruption (She et al., 1998; Offermann et al., 2010; Keckhut et al., 1995; Thulasiraman and Nee, 2002; Wallis et al., 2022). However, these estimates have large uncertainties regarding their size and timing. There are few studies that use an atmospheric model with a fully resolved mesosphere to simulate the temperature and wind field changes after a volcanic eruption. Rind et al. (1992) used the Goddard Institute for Space Studies (GISS) Global Climate/Middle Atmosphere

model and showed annually averaged results up to 90 km altitude. Their study indicated a warming of the upper mesosphere of both hemispheres that was attributed to circulation changes. They noticed a decrease of the Eliassen-Palm flux in the upper mesosphere and concluded that this indicates a reduced divergence of vertical sensible heat flux in this region which would explain the heating of the upper mesosphere. Ramesh et al. (2020) simulated the variability of the middle atmosphere from 1850 to 2014 with the Whole Atmosphere Community Climate Model (WACCM) and analyzed, besides other forcings, also

the response to volcanic aerosol. Their results show a significant and positive temperature anomaly in the summer mesopause region (see their Figure 7), however, they did not discuss this finding further.

Since volcanic eruptions have a significant impact on the stratosphere it is plausible that this perturbation also affects the mesosphere as these atmospheric layers are dynamically coupled. Depending on the background conditions, gravity waves can propagate from the troposphere up to the mesosphere and locally distribute energy by wave breaking. The mesosphere exhibits a

summer-to-winter-pole circulation (responsible for the low temperatures of the polar summer mesopause) that is mainly driven by momentum deposition from gravity waves. This forcing depends largely on the circulation of the atmosphere below as this circulation acts to filter part of the propagating waves (Körnich and Becker, 2010). Additionally to this dominantly vertical coupling, dynamical processes can also cause interhemispheric coupling. An impact of the circulation in the winter stratosphere on the summer mesosphere has been both observed in satellite data (Karlsson et al., 2007) and described theoretically (Becker

and Fritts, 2006; Körnich and Becker, 2010; Karlsson and Becker, 2016; Smith et al., 2020).

The focus of this study is the impact of volcanic eruptions on the mesosphere, a region of the atmosphere that is not as well observed and documented as the stratosphere below. We aim at identifying the potential dynamical mechanisms that drive changes in this region due to volcanic perturbations. For this purpose, we use the upper-atmosphere icosahedral non-hydrostatic

(UA-ICON) model to simulate the atmospheric response to an idealized strong volcanic eruption that injects about 20 Tg S
into the atmosphere, which is about twice as much as the 1991 Pinatubo eruption.

This paper is structured as follows. The UA-ICON model and the experiment runs will be introduced in Section 2. Section 3 will present the main results that are afterwards discussed in Section 4. The paper ends with conclusions given in Section 5.

## 2   Tools and experiments

### 2.1   UA-ICON model

UA-ICON is an extension of the ICOsahedral Non-hydrostatic (ICON) general circulation model for the upper atmosphere (Borchert et al., 2019). The ICON model was jointly developed by the Max Planck Institute for Meteorology and the German weather service (Zängl et al., 2015). The upper atmosphere extension expands the dynamical core from shallow to deep atmosphere dynamics. The shallow approximation neglects terms related to the spherical curvature of the atmosphere as well as the variation of the gravitational field. These terms are modified in order to account for a more realistic deep atmosphere (Borchert et al., 2019). Furthermore, a physics package for the upper atmosphere is implemented. This study uses a R02B04 grid, i.e., a triangular grid with about 160 km horizontal mesh size, with 120 vertical levels and a model top at 150 km altitude. Each run is initialized from an Integrated Forecasting System (IFS) analysis provided by the European Centre for Medium-Range Weather Forecast (ECMWF). Our first ensemble member uses the IFS products starting in the year 1980, the second ensemble member starts in 1981 etc. The IFS model has a model top at approximately 0.01 hPa and is used to initialize the state of the lower and middle atmosphere in UA-ICON. Above, vertical temperature profiles from Bates and Massey (1959), Hedin (1983) and Fleming et al. (1988) are used to initialize the upper atmosphere. The UA-ICON model does not simulate atmospheric chemical reactions, but prescribed climatological fields of $O_3$, $O_2$, $O$, $NO$, $CO_2$, $CH_4$ and $N_2O$ are used in the radiative transfer calculations. Additionally, climatologies are prescribed for tropospheric aerosol optical properties, the sea surface temperature and sea ice. The air masses in the upper atmosphere are no longer in local thermodynamic equilibrium and modifications included in the physics package are necessary. Non-local thermodynamic equilibrium cooling of $CO_2$ and ozone are implemented above 65 km using Fomichev and Blanchet (1995) (with modification from Fomichev et al. (1998)), the $CO_2$ absorption above 19.25 km using Ogibalov and Fomichev (2003) and the $NO$ cooling above 60 km by Kockarts (1980). A 35-year HAMMONIA simulation provides the monthly and zonal mean chemical heating rates that are used in UA-ICON. Spectral solar irradiance is prescribed at the model top. UA-ICON uses the PSrad radiation package (Pincus and Stevens, 2013) (covering wavelengths longer than 200 nm) as the standard ICON model, but additionally implements the UA package. Above 50 km, this package uses a model by Strobel (1978) to account for the $O_2$ Schumann-Runge bands from 175 to 205 nm and the continuum from 125 to 175 nm. Above 90 km, a model based on Richards et al. (1994) is used to include the extreme UV range from 5 to 105 nm. A validation of UA-ICON against the Sounding of the Atmosphere using Broadband Emission Radiometry (SABER) on the Thermosphere Ionosphere Mesosphere Energetics Dynamics (TIMED) satellite and against a climatology from Upper Atmosphere Research Satellite Reference Atmosphere Project (URAP) is given in Borchert et al. (2019). The Doppler Spread Parameterization of

Hines (1997) is used to implement the momentum propagation and deposition of sub-grid scale non-orographic gravity waves. Effects of sub-grid scale orography, including orographic gravity waves are described following Lott (1999).

## 2.2 Volcanic experiments

A volcanic forcing representative of a tropical volcanic injection of 20 Tg S into the lower stratosphere was generated offline by the Easy Volcanic Aerosol (EVA) forcing generator which is described in detail by Toohey et al. (2016). This relatively large amount of approximately twice of what was emitted in the 1991 Pinatubo eruption (Guo et al., 2004) was chosen to obtain a large signal-to-noise ratio. The EVA module determines optical properties (aerosol extinction, asymmetry parameter and single scattering albedo) of an idealized stratospheric volcanic aerosol distribution which itself is calculated in a simple three-box model using eruption location, and date, and amount of injected sulfur as input parameters. The model was originally tuned to represent the distribution resulting from the Pinatubo eruption. Figure 1a shows the aerosol optical depth used to calculate the radiative transfer in the visible spectral range between 0.44 and 0.63 $\mu$m. The AOD in the visible range is important to determine the radiative forcing and, hence, the climate response. The extinction coefficient between 8.47 and 9.26 $\mu$m (infrared range) for the first post-eruption November is shown in Figure 1b and provides insight into aerosol extinction of mostly terrestrial wavelengths that are responsible for heating the lower stratosphere (Arfeuille et al., 2013), as discussed below. Two experiments were performed (Vol1 and Vol2), each containing 10 ensemble members that simulate 27 months after the eruption event. The volcanic runs in Vol1 and Vol2 were all started in May and the eruption was simulated on June 15 as a reference to the Pinatubo eruption. The experiments only differ in the parameterization of sub-grid scale orographic (SSO) effects. We additionally performed non-volcanic ensemble simulations as reference for each of the two experiments (Ref1 and Ref2).

The UA-ICON configuration used here calculates SSO effects according to the parameterization by Lott (1999). We used two different settings of the Lott (1999) parameterization of sub-grid scale orographic effects chosen in order to have experiments with different representations of the polar vortex and test the robustness of the mesospheric response to volcanic aerosol under different reference circulation states. The parameterization represents two different effects of unresolved orography: a) low-level blocking of near-surface flow that would be forced to flow around the orographic barrier. The strength of his effect can be scaled by the tuning parameter $C_d$. b) Momentum transfer by gravity waves caused by the orography (scaled by parameter G). The first experiment follows the setting of Borchert et al. (2019) with G=0.1 and $C_d$=0.01, while in the second experiment, G=0.05 and $C_d$=0.2 are used. This means that the effects of atmospheric drag from gravity waves expected from unresolved orography is reduced and additionally the low-level blocking effect of unresolved orography is enhanced. Therefore, the second experiment explicitly reduces the contribution of orographic gravity wave generation.

The model output was interpolated from an irregular ICON grid to a regular (t64) grid and fixed geometric height levels. We also performed a transformed Eulerian mean (TEM) diagnostic (Hardiman et al., 2010) in order to determine the Eliassen-Palm flux divergence and the residual stream function. TEM calculations were performed for each time step of the 3-hourly model output and then averaged monthly.

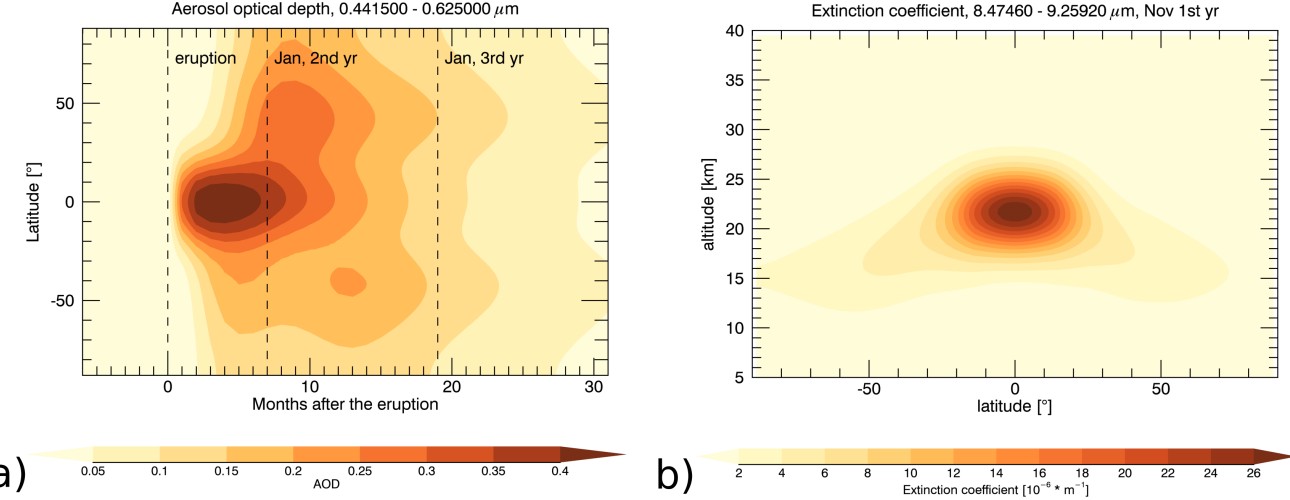

**Figure 1.** a) Aerosol optical depth (AOD) between 0.44 and 0.63 $\mu$m and b) extinction coefficient between 8.47 and 9.26 $\mu$m for the first post-eruption November.

## 3 Results

We would like to focus on the impact on the summer hemisphere half a year after the eruption. This region is especially interesting, because the very cold summer mesopause allows for the formation of noctilucent clouds. Their properties, such as occurrence frequency and brightness, are sensitive to temperature variability in the mesopause (Baumgarten et al., 2010; Pertsev et al., 2014; von Savigny et al., 2020a) and are potentially affected by any temperature perturbation in this region.

As expected, the simulations clearly show that a strong tropical eruption affects the temperature of the middle atmosphere.
Figure 2a shows the ensemble mean reference temperature and Figure 2b the temperature anomaly (Vol1 - Ref1) for the middle atmosphere in November, i.e. five months after the simulated eruption. Temperatures simulated in Ref1 for the summer mesopause between 80 and 90 km reach minimum values as low as $\approx$ 145 K in November and even $\approx$ 120 K in December. The volcanic eruption in Vol1 causes a strong positive temperature anomaly in the tropical lower stratosphere of up to 16 K (Figure 2b) that is mostly due to the absorption of longwave radiation by volcanic sulfate aerosol. This temperature anomaly
decreases only slightly to 14 K in February (2e). The cooling of the tropical stratosphere above is due to increased upwelling and adiabatic cooling related to an acceleration of the Brewer-Dobson circulation. This is apparent in Figure 3 where the clockwise circulation between 20 and 40 km in the northern hemisphere (NH) is strengthened during November to February. Here, the residual mass streamfunction $\chi$ is calculated as

$$\chi = A cos\phi \int \bar{v}^* \rho \, dz \tag{1}$$

with A being the radius of the Earth, $\phi$ the latitude, $\bar{v}^*$ the residual meridional velocity and $\rho$ the density. This circulation anomaly also results in a warming of the winter polar lower stratosphere where the downwelling of air masses increases. A

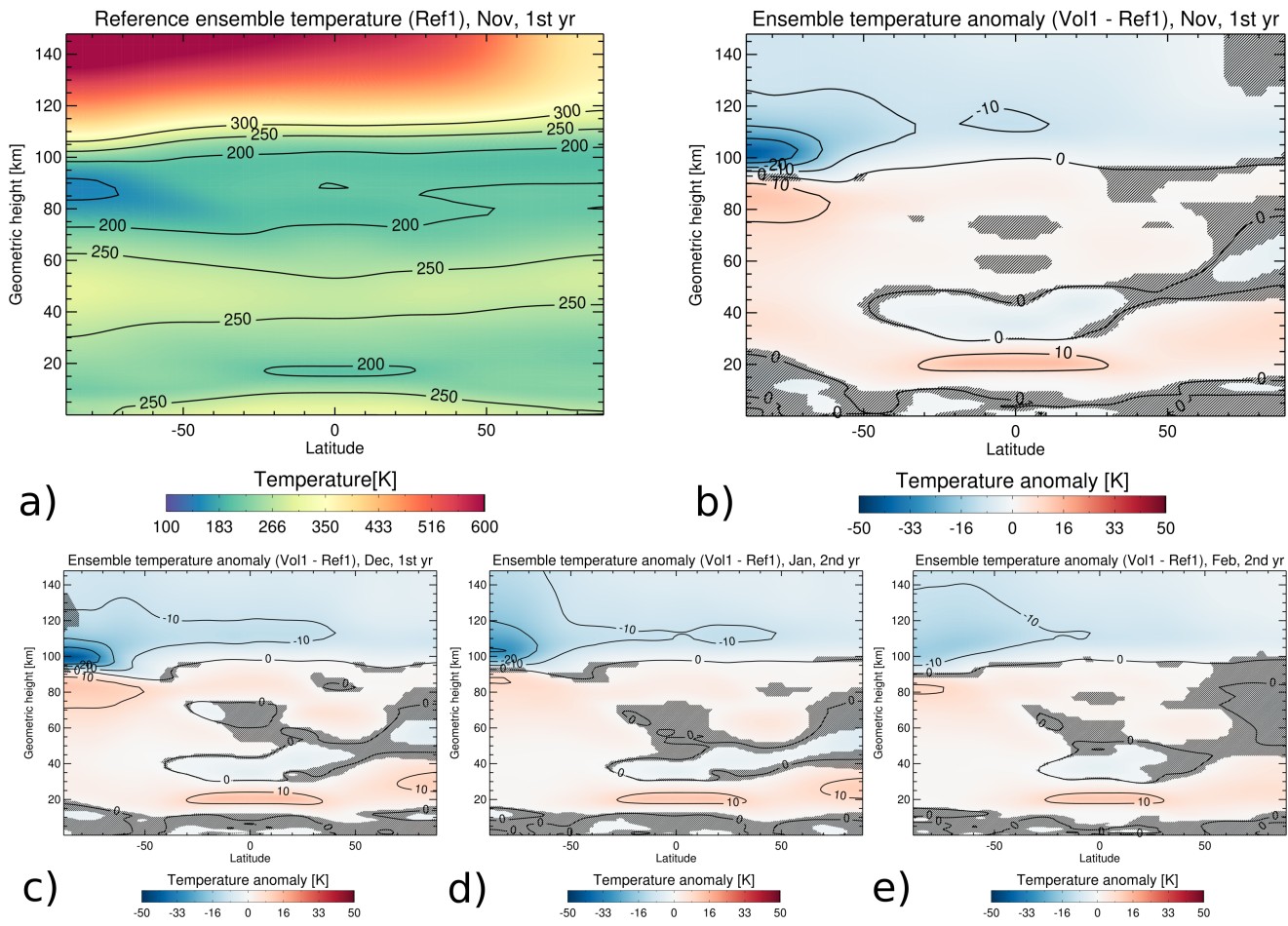

**Figure 2.** a) Zonal, monthly and ensemble mean reference temperatures (Ref1) for the first November after the eruption. Zonal, monthly and ensemble mean temperature anomalies (Vol1 - Ref1) for b) November, c) December, d) January and e) February. Hatched areas are not significant at a 95% confidence interval using a Student's t-test.

small area of cooling below the polar stratospheric warming was also simulated in other studies and is interpreted as a result of radiative longwave cooling (Toohey et al., 2014).

We will now discuss the potential dynamical mechanism that causes the simulated warming of the polar summer mesopause described above. It is apparent from the temperature anomalies presented in Figure 2b-e that the volcanic heating of the tropical lower stratosphere results in an altered stratospheric temperature gradient between the tropics and the southern hemispheric (SH) high latitudes. The thermal wind balance alters the zonal mean zonal wind in this region, which is apparent in the ensemble zonal wind anomaly (Vol1 - Ref1) in Figure 4b-e. Here, the zonal mean easterlies prevailing in the summer middle atmosphere between about 20 and 70 km are reduced by up to 20 m/s in December (and already up to nearly 13 m/s in November). The

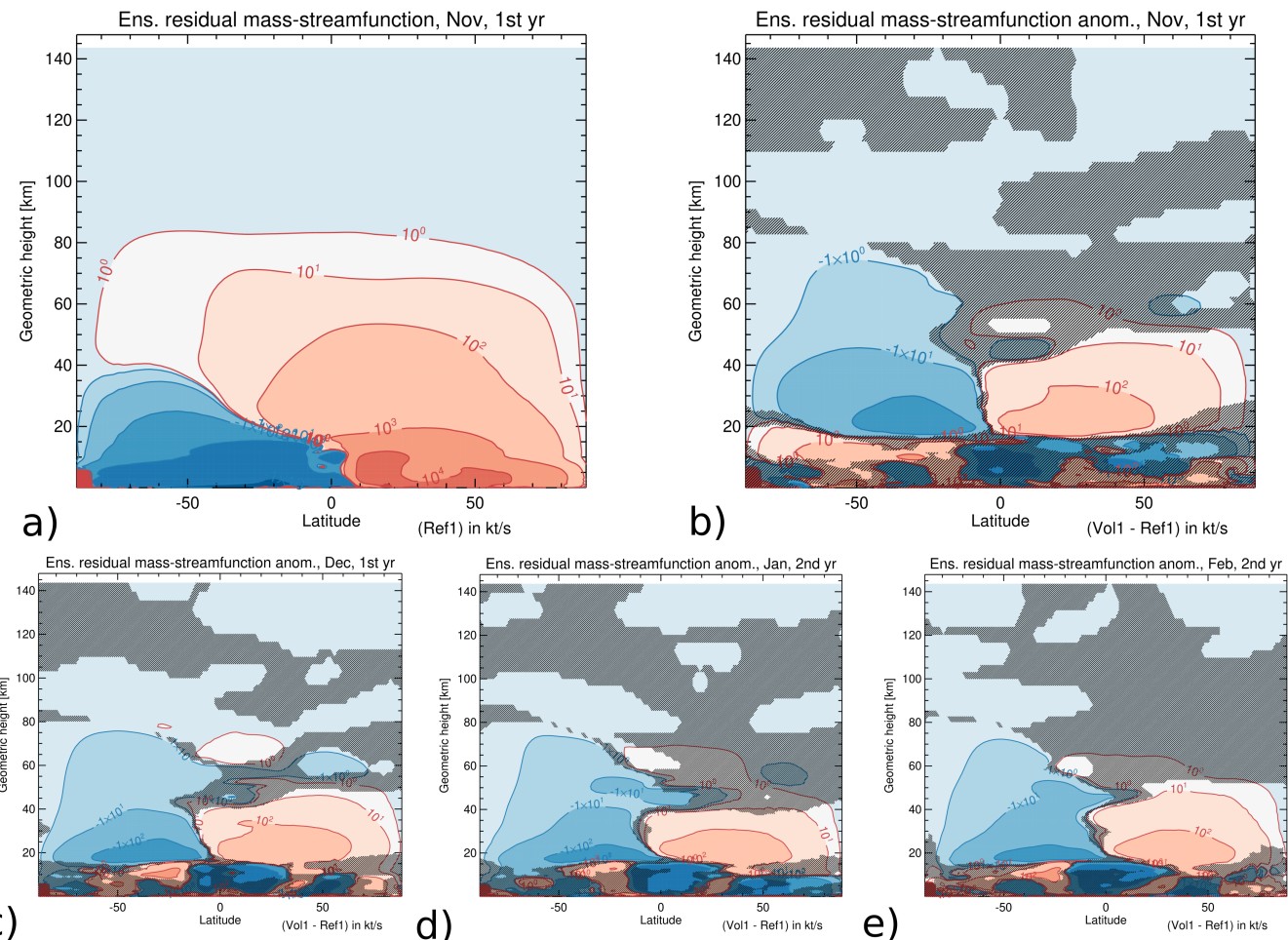

**Figure 3.** Similar to Figure 2, but for the zonal, monthly and ensemble mean residual mass-streamfunction in units of kt/s. The large variability above the south pole up to 10 km altitudes is an artifact from the interpolation onto fixed geometric height levels.

winter vortex response in Figure 4b-e is, however, unexpected as the positive zonal wind anomalies between 50 – 90 °N and 30 – 60 km indicates a polar vortex weakening instead of a strengthening. Although there is a strengthening of the westerlies at 30 °N, i.e. where the anomalous heating rate gradient is strongest, this does not result in an enhancement of the westerlies of the polar vortex as was suggested in a former study (Bittner et al., 2016). We will address the polar vortex response further below in our second experiment (Ref2 and Vol2).

The altered zonal winds affect the filtering of the gravity waves in the summer hemisphere, i.e waves with strongly easterly phase speeds can propagate upwards to the mesosphere. This is suggested by the negative anomaly of the zonal wind tendency (Vol1 - Ref1), i.e. the non-orographic gravity wave drag in zonal direction. It reaches up to ≈ -37 m/s/day in November as can be seen in Figure 5b before weakening in February (Figure 5c-e). The gravity wave drag anomaly forms a dipole with a positive anomaly above the negative. The zero anomaly line is simulated at approximately 95 km near the pole and bends downward

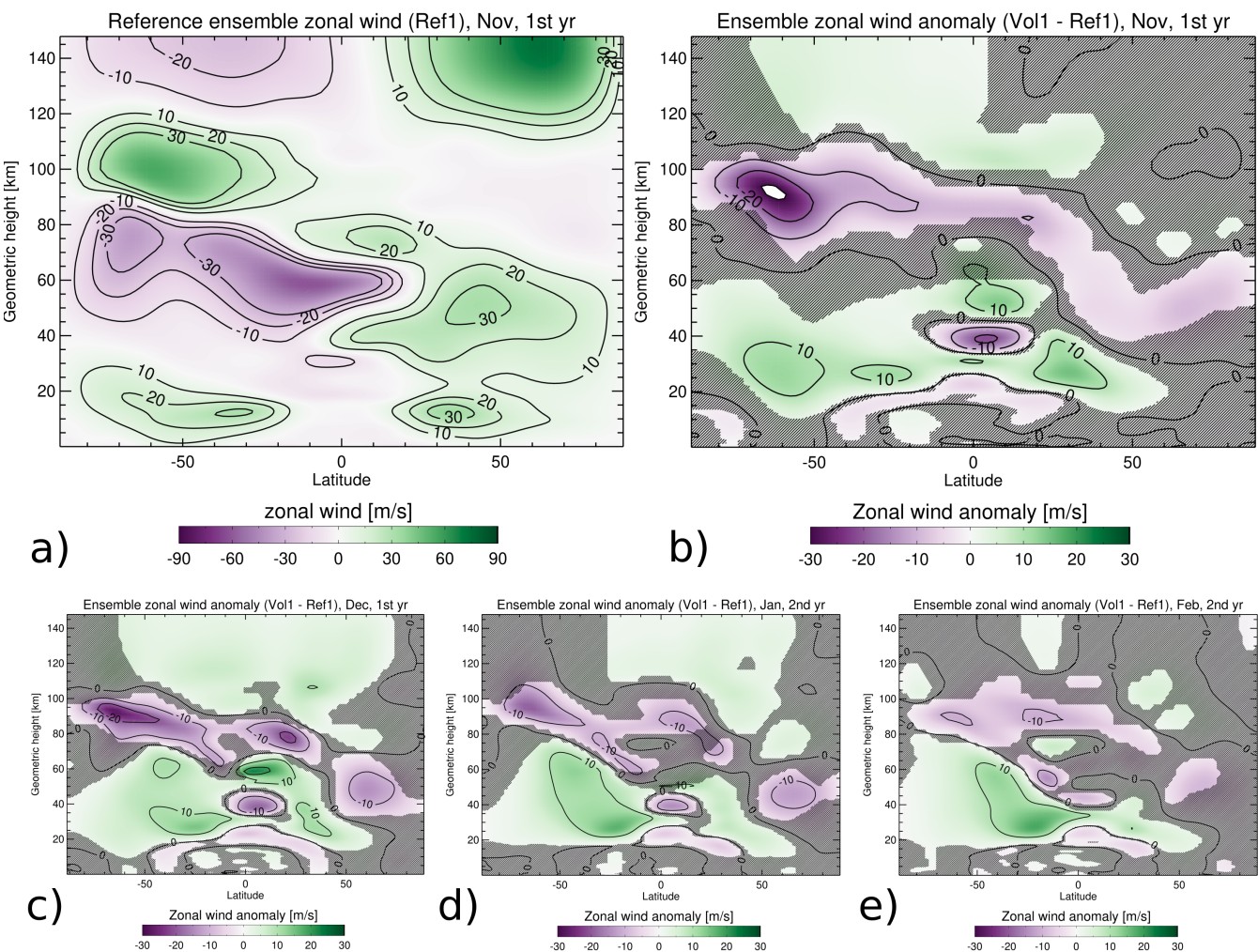

**Figure 4.** Similar to Figure 2, but for the zonal, monthly and ensemble mean zonal wind.

towards the equator. This positive/negative dipole indicates an upward shift of the gravity wave breaking levels caused by the altered zonal winds in the summer stratosphere.

The existence of a negative gravity wave drag anomaly indicates that the upwelling of air over the summer polar region is reduced. It also reduces the adiabatic cooling in this area. This can be clearly seen as a positive temperature anomaly in the summer mesosphere of up to 15 K in November (Figure 2b). This warming is strongest in November and December (in this month up to 14 K), but still apparent until February (Figure 2b). A comparison with the temperature state of the reference run (Ref1) confirms that the warming also includes the polar summer mesopause region (Figure 2a) and an increase in the altitude of the mesospheric cold point indicates that the mesopause is moving upwards in Vol1 (Figure S1). The maximum of the summer mesopause warming for latitudes polewards of approximately 70°S is reached already in November (Figure 6)

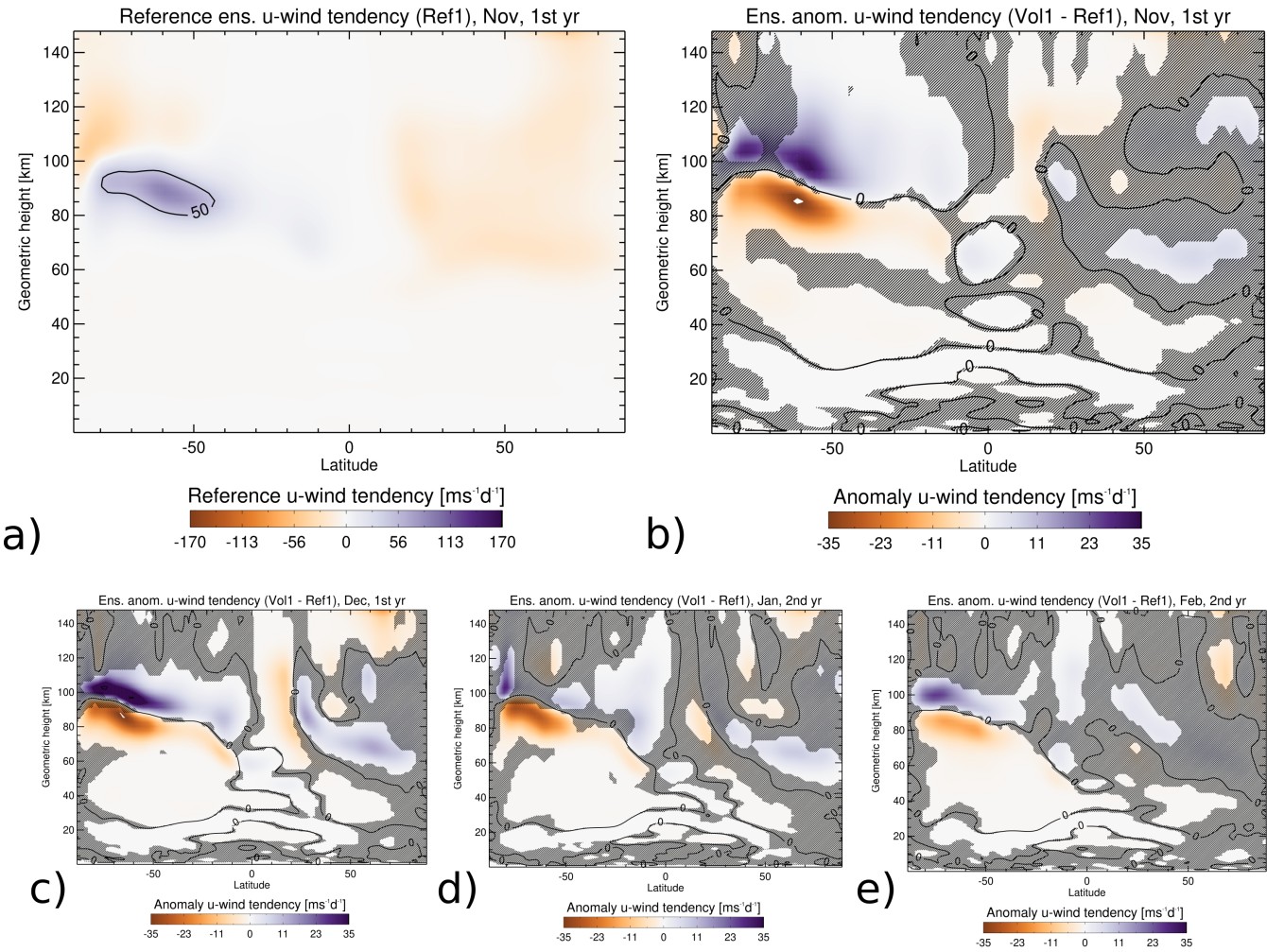

**Figure 5.** Similar to Figure 2, but for the zonal, monthly and ensemble mean zonal wind tendency (i.e. the non-orographic gravity wave drag in zonal direction).

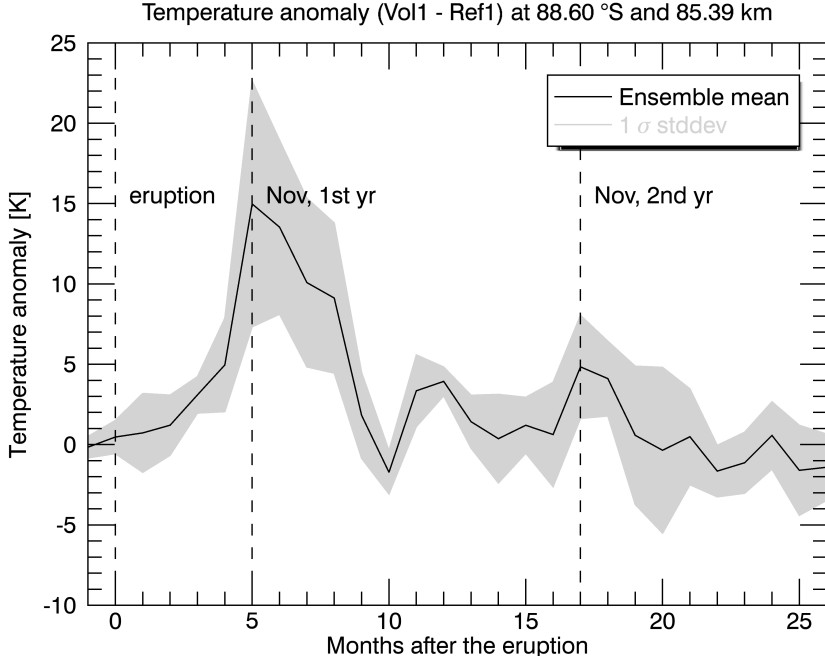

**Figure 6.** Time series of zonal, monthly and ensemble mean temperature anomalies (Vol1 - Ref1) at 88.6°S and 85.39 km altitude. Gray shading indicates 1 standard deviation of the 10 ensemble member anomalies (Vol1 - Ref1). The date of the eruption as well as of the first and second post-eruption Novembers are marked with dash lines.

and between approximately 50°S-70°S one month later in December. There is also a mesopause warming in the second austral
summer after the eruption, but it is much weaker and not significant in January (not shown).

Besides the parameterized gravity waves discussed above, the model resolves planetary waves that can also affect the mean flow when they break and deposit their momentum. We want to test the hypothesis that the major effect for the polar summer mesopause warming is caused by gravity waves and not the resolved planetary waves. The resolved waves exert a forcing on the mean flow that can be visualized by the divergence of the Eliassen-Palm (EP) flux (Andrews, 1987). Figure 7a shows
the ensemble mean reference (Ref1) EP flux divergence. In the summer mesopause region, the EP flux divergence is positive between 80 – 100 km bending downwards towards the equator and negative above up to 120 km between 20 – 90°S. The EP flux divergence anomalies (Vol1 - Ref1) in this region have a reversed sign (Figure 7b), i.e. a positive anomaly at 100 km bending downwards is situated above a negative anomaly, although the latter is not significant everywhere. The dipole patterns of the gravity-wave drag anomaly (Figure 5b) and the EP flux divergence anomaly (Figure 7b) are qualitatively similar in the
summer mesopause region but the latter is weaker and less significant. Figures 5 and 7 therefore indicate that the impact of the gravity waves dominates the volcanic signal in the circulation and temperature of the summer mesopause region.

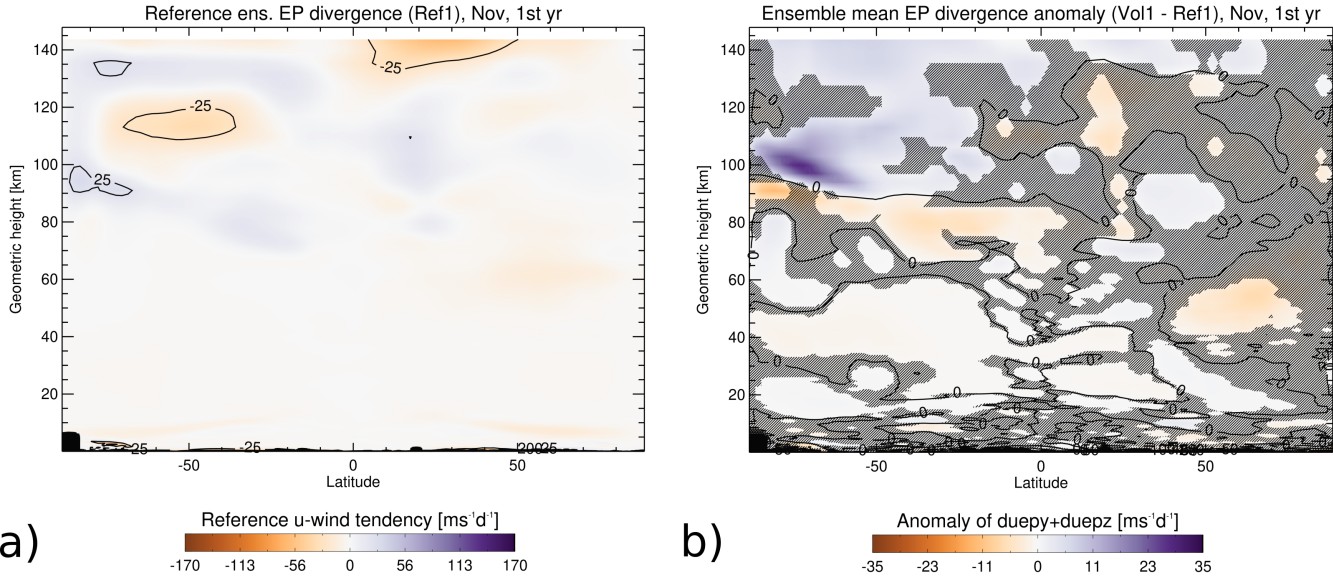

**Figure 7.** a) Zonal, monthly and ensemble mean of the EP flux divergence reference (Ref1) and b) anomaly (Vol1 - Ref1) for the first post-eruption November. Hatched areas are not significant at a 95% confidence interval using a Student's t-test. The large variability above the south pole up to 10 km altitudes is an artifact from the interpolation onto fixed geometric height levels.

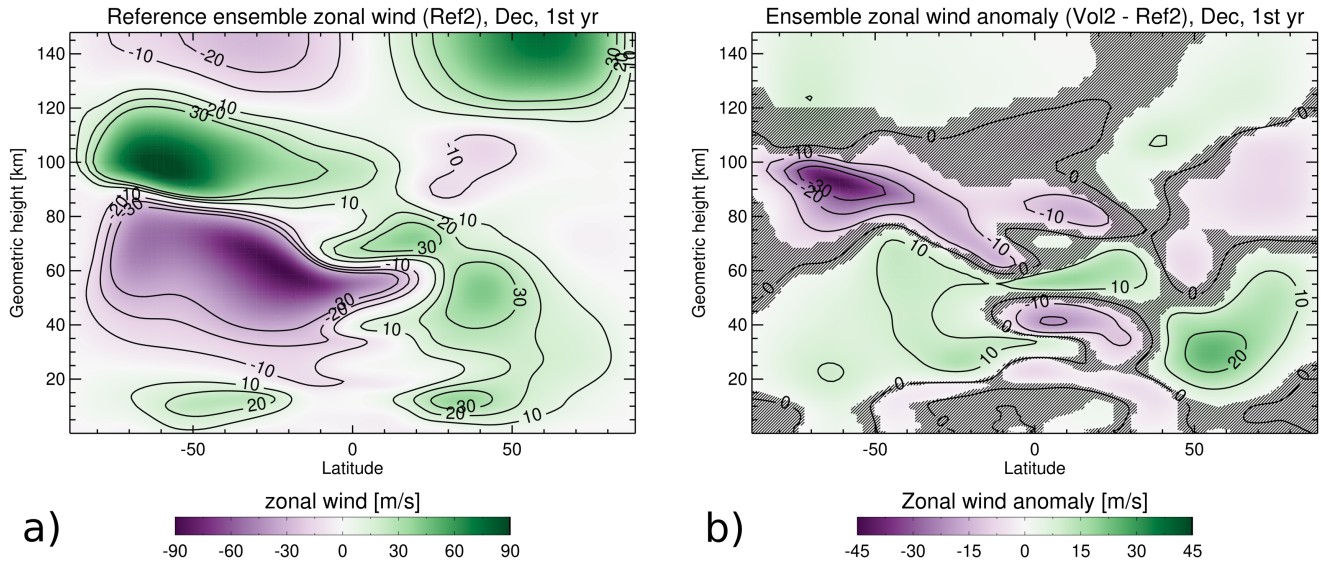

**Figure 8.** a) Zonal, monthly and ensemble mean of the zonal wind reference (Ref2) and b) anomaly (Vol2 - Ref2) for the first post-eruption December. Hatched areas are not significant at a 95% confidence interval using a Student's t-test.

While the vertical coupling in the summer hemisphere provides a plausible explanation for the warming signal in the summer mesopause, it is unclear if interhemispheric coupling may contribute to it. Additionally, the simulated weakening response of the polar vortex in the winter stratosphere seems unrealistic when it is compared to the typical response of GCMs to aerosols from strong tropical volcanic eruptions (Bittner et al., 2016). As a consequence, potential interhemispheric coupling could also lead to an unrealistic mesospheric response to volcanic aerosol. For the second experiment (Ref2 and Vol2), the orographic gravity wave parameterization was therefore changed with the intention to strengthen the polar vortex in the reference simulation and to create a different, and hopefully more realistic, response of the winter stratosphere and thereby test the robustness of the mesospheric response under different stratospheric responses. Indeed, this changed tuning has a strong impact on the zonal mean zonal wind anomaly in the winter stratosphere (Vol2 - Ref2). Positive anomalies of up to $\approx$ 15 m/s in November (not shown) and up to 25 m/s in December (Figure 8b) at 60°N and 30 km indicate a strengthening of the NH polar vortex.

However, the response of the mesosphere to volcanic forcing is qualitatively similar in experiments 1 and 2. The altered gravity wave parameterization results in a significant positive temperature anomaly of up to 21 K in the polar summer mesopause region from 80 - 95 km altitude and between 70°S - 90°S in November (Vol2 - Ref2, see Figure 9b). A warming of up to nearly 10 K is still present in February (Figure 9e). We compare the results of the two sets of simulations with different gravity wave parameters by showing the differences between the temperature anomaly response in both experiments (i.e. (Vol2 - Ref2)-(Vol1 - Ref1)) in December in Figure 10a. The NH lower stratosphere up to 40 km at high latitudes shows a negative temperature anomaly difference with a positive anomaly difference above (between 40 to 80 km). A negative temperature anomaly difference is found above the equator in the mesosphere at 80 km and also in the SH polar atmosphere at 100 km. This distinct pattern is similar to temperature anomaly patterns known from interhemispheric coupling, i.e. a coupling existing between the winter stratosphere and the summer mesosphere. This might hint at a similar phenomenon in our study, although the temperature anomaly differences are not significant everywhere. This proposed mechanism is further supported by the difference in the anomaly of the zonal wind tendency shown in Figure 10b. There is a significant negative anomaly difference between 40-90°N and 40 to 100 km that is probably due to a change in the gravity wave filtering (less eastward propagating gravity waves can enter the mesosphere (Karlsson et al., 2009)) as a consequence of the altered meridional temperature gradient (Figure 10a) and hence zonal wind (not shown) in the stratosphere between both experiments. A positive zonal wind tendency anomaly difference is simulated in the polar summer atmosphere at 100 km and a negative anomaly difference below. The negative anomaly difference below is not significant in most areas, but the apparent dipole structure could hint at a vertical displacement of the gravity wave drag. This pattern is similar to the previous study on interhemispheric coupling by Karlsson et al. (2009), although these authors discussed anomalies and not anomaly differences. The vertical displacement of the gravity wave drag is probably caused by the zonal wind changes in the summer hemisphere (not shown) due the altered meridional temperature gradient between the equatorial mesosphere (negative temperature anomaly difference) and the polar mesosphere. This seems to change the height of the gravity wave drag in the mesopause region as gravity waves with easterly phase speeds break at lower altitude and gravity waves with westerly phase speeds at higher (Karlsson et al., 2009). The parametrisation of the second experiment (Ref2 and Vol2) also alters the residual circulation in the atmosphere. Figure 10c shows the difference of the ensemble mean residual mass-circulation anomalies between both experiments ((Vol2 - Ref2)-(Vol1 - Ref1)). The negative anomaly difference

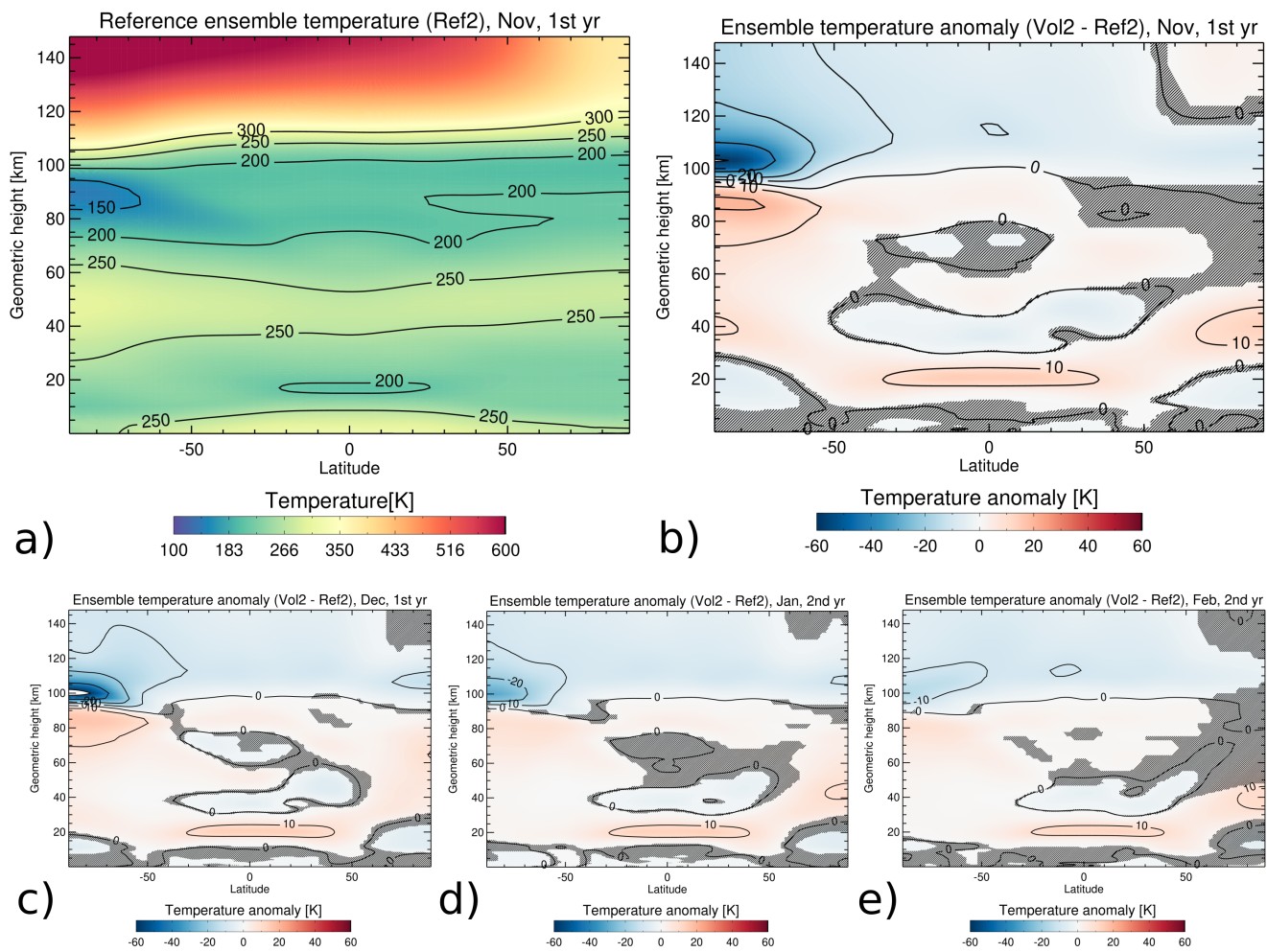

**Figure 9.** a) Zonal, monthly and ensemble mean reference temperatures (Ref2) for the first November after the eruption. Zonal, monthly and ensemble mean temperature anomalies (Vol2 - Ref2) for b) November, c) December, d) January and e) February. Hatched areas are not significant at a 95% confidence interval using a Student's t-test.

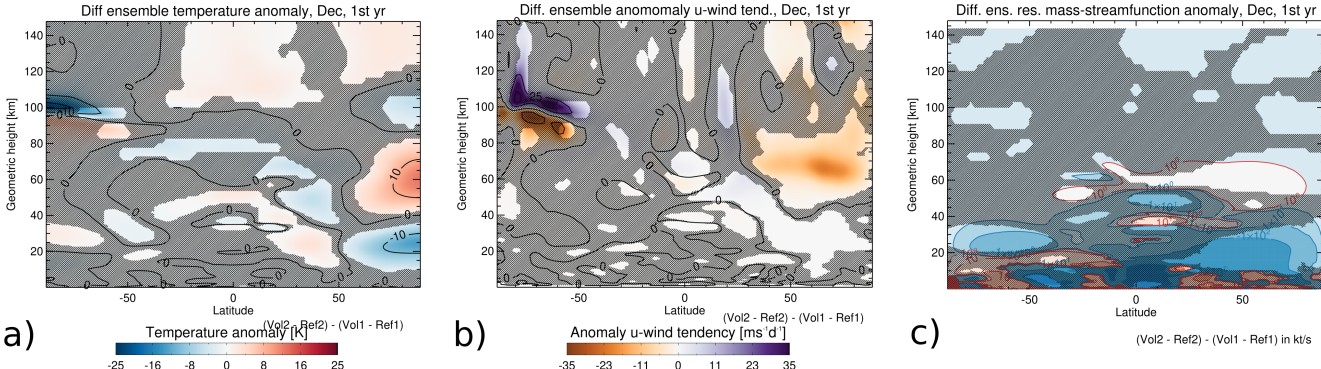

**Figure 10.** Difference of the zonal, monthly and ensemble mean anomalies ((Vol2 - Ref2)-(Vol1 - Ref1)) for a) the temperature, b) the zonal wind tendency (i.e. gravity wave drag) and c) the residual mass-streamfunction for the first post-eruption December. Hatched areas are not significant at a 95% confidence interval using a Student's t-test.

(counter-clockwise circulation) between 10 to 40 km indicates that the lower branch of the Brewer-Dobson circulation in the NH is weakened.

It is difficult to separate the contributions of intra- and potential interhemispheric coupling as the latter mechanism might
be present in both experiments. The temperature response of the summer mesosphere is, however, qualitatively similar in both experiments despite the winter stratospheric state being very different. This indicates that interhemispheric coupling, if such a mechanism occurs in our study, would have only a relatively small contribution to the overall dynamical mechanism.

## 4 Discussion

The model simulations indicate that the perturbation due to the stratospheric aerosol changes the temperature of the polar
summer mesosphere during the first post-eruption austral summer. This temperature response seems to be dominated by intra-hemispheric coupling. The weakened easterlies in the stratospheric summer hemisphere allow the propagation of more gravity waves with easterly phase speed into the mesosphere (Körnich and Becker, 2010). Their breaking reduces the net westerly gravity wave momentum deposition in the upper mesosphere which consequently reduces the high latitude upwelling and increases mesopause temperatures. The transition to westerlies is shifted upwards and with it the region where critical line breaking
of gravity waves with westerly phase speed may occur. This explains the vertical dipole of easterly and westerly zonal wind acceleration by gravity wave drag seen in Figure 5b-e. The variability of stratospheric zonal winds and their effect on the filtering of gravity waves is also discussed in the literature. This mechanism was for example proposed to explain the correlation between the date of the SH winter-to-summer transition and the onset of the noctilucent cloud season by Gumbel and Karlsson (2011). A shifting (although downwards in this case) of both the gravity wave breaking level and the mesospheric residual
circulation was also found in simulations of the dynamical effects of solar proton events on the middle atmosphere (Becker

and von Savigny, 2010). Lübken et al. (2015) used intrahemispheric coupling to explain the observation of an unusually cold elevated summer mesopause over Davis (Antarctica) around summer solstice in Fe lidar data.

A possible contribution from interhemispheric coupling is hinted at by comparing both experiments presented in this study. The gravity wave parameterization was altered for the second experiment (Ref2 and Vol2) and resulting in a strengthening of the NH polar vortex for the eruption runs (Vol2). The winter stratosphere generally exhibits a strong interannual variability and as we only generate 10 ensemble members for Vol2 we unfortunately cannot state if this is a robust response (Azoulay et al., 2021). We nevertheless regard this as a chance to investigate interhemispheric coupling, because the difference between the temperature anomalies of both experiments shows a pattern that is associated with this phenomenon (Karlsson et al., 2009; Körnich and Becker, 2010; Karlsson and Becker, 2016). This mechanism suggests that a strengthened polar vortex in the winter hemisphere in the second experiment (due to the decrease in gravity wave source and the increase in low-level flow blocking) reduces the Brewer-Dobson circulation (as suggested by Figure 10c). Essentially, this decreases the downwelling in the winter polar stratosphere leading to a cooling of this region, while a reduced upwelling causes anomalous warming in the tropical stratosphere. The enhanced winter polar vortex can filter out more gravity waves with an eastward phase speed that do not reach the mesosphere. Overall, a negative gravity wave drag anomaly is simulated in the winter mesosphere with an increase in downwelling and associated adiabatic heating in the polar winter mesosphere. The tropical mesosphere exhibits a cooling due to this altered residual circulation. The temperature anomalies in the tropics alter the zonal mean zonal winds in the summer hemisphere and change the altitude of gravity wave breaking. Karlsson et al. (2009) found a dipole structure in the gravity wave drag response over the summer polar mesopause region simulated in their Canadian Middle Atmosphere Model. Their simulation showed a positive gravity wave drag anomaly over a negative anomaly which they interpreted as a vertical shift of the gravity wave drag. This is also found in the UA-ICON simulations comparing anomaly differences of two experiments with different SSO parameterization, although most of the negative gravity wave drag anomaly difference is not significant in most parts of the summer mesopause region (Figure 10b). Finally, the changed residual circulation in the mesosphere induces a negative temperature anomaly difference that is located above the polar summer meopause in the UA-ICON simulation. A positive polar summer mesopause temperature is found in the anomaly difference between both experiments ((Vol2 - Ref2) - (Vol1 - Ref1)), however, it is mostly not significant (Figure 10a). Moreover, a change of the sub-grid scale orographic parameterization is not only affecting the volcanic perturbation, but also leads to differences in the volcanic-free reference runs. Interhemispheric coupling is possibly involved in both experiments individually. However, the polar summer mesopause temperature response is similar for both experiments despite the clear differences in the winter stratosphere. A potential interhemispheric coupling therefore seems to be less relevant for the middle atmospheric response to a tropical volcanic eruption in our simulation compared to the dominant intrahemispheric coupling.

Our simulations indicate that intra- and interhemispheric coupling result in a positive temperature anomaly of the summer mesopause just 5 to 6 months after the simulated eruption. Observation of the mesospheric temperature response after strong volcanic eruptions are sparse because of a lack of such events during the satellite era. Some studies observed a mesospheric warming after the 1991 Pinatubo eruption, but reported different delays for this perturbation. These observations can only be qualitatively compared to our model results (in the following we will only discuss Ref1 and Vol1), because the experiments

assume twice as much $SO_2$ emission as was released during the Pinatubo eruption. An almost instant response with maximum temperature anomalies measured in 1991 was reported for OH* rotational temperatures over Wuppertal at 51.3° N (Kalicinsky et al., 2016; Offermann et al., 2010) and for satellite-based HALOE temperature measurements (Wallis et al., 2022). UA-ICON does not simulate a warming of the upper mesopause at 50°N, as was observed in OH* temperature measurements. The standard deviation of the ensemble members 2 - 3 months after the simulated eruption is, however, relatively high among the members (not shown). Nevertheless, UA-ICON simulates a peak in the temperature anomaly in the first post-eruption December at 85.4 km and between 15-25 °N similar to the analysis of HALOE temperature data in Wallis et al. (2022) (Figure S2a). Both the ground-based Rayleigh lidar at 44° N (Keckhut et al., 1995) and the satellite instrument High Resolution Doppler Image (Thulasiraman and Nee, 2002) reported a warming in both 1992 and 1993. Depending on the altitude, some of these observations are also apparent in UA-ICON. The model simulates a positive temperature anomaly in the first post-eruption summer (analogous to the summer of 1992) at 78 km for 45°N in agreement with Rayleigh lidar data, but in contrast did not simulate a warming in the second post-eruption summer (Figure S2b). Focusing on the winters, UA-ICON simulates a warming in the first and second post-eruption winter in 66 km altitude at 45°N (Figure S2c). Na lidar data over Fort Collins at 40.6° N (She et al., 1998) indicate a mesopause temperature peak as late as 1.5 year after the eruption, i.e. in 1993. At 41°N and 100 km altitude, UA-ICON simulates a small peak in temperature anomaly 1.5 years after the simulated eruption with a large standard deviation that could hint at a possible temperature anomaly maximum in 1993 due to a combination of volcanic perturbation and internal variability as was observed in the lidar data over Fort Collins (Figure S2d).

The UA-ICON model does not calculate atmospheric chemistry interactively, e.g. it does not account for heterogenous chemical reactions on volcanic sulfate aerosols. In the atmosphere, those reactions could deplete nitrogen dioxide (Aquila et al., 2013), activate chlorine (Solomon, 1999) and hence change the ozone concentration in the stratosphere (Rozanov et al., 2002). Ozone is important for the radiative balance in the atmosphere and changes in its concentration could affect the temperature and subsequently the atmospheric circulation. Rozanov et al. (2002) used the University of Illinois at Urbana-Champaign (UIUC) stratosphere–troposphere general circulation model to separate the volcanic impact of radiative aerosol heating and ozone depletion due to heterogenous chemical reaction on sulfate aerosols. They found that the temperature anomaly in the lower stratosphere is predominantly caused by the radiative heating of the aerosols and not the relatively small cooling by volcanic ozone depletion. These results are supported by Kilian et al. (2020) in their chemistry-climate model EMAC. The warming of the tropical lower stratosphere itself will probably alter the transport of ozone and in turn have an impact on the temperature state of the atmosphere. UA-ICON is run with a prescribed climatological field for the concentration of ozone. Hence, the omission of sulfate aerosol interaction with ozone and also the neglect of chemical transport as a result of changed atmospheric circulations are a clear limitation of this model. We would argue, however, that the qualitative argument of our study is still valuable to explain the dynamic response of the atmosphere due to volcanic eruption.

## 5 Conclusions

The upper-atmosphere icosahedral non-hydrostatic model UA-ICON was used to simulate the atmospheric response to a tropical volcanic eruption emitting about twice as much sulfur into the atmosphere as the 1991 Pinatubo eruption. As expected, a heating of the tropical lower stratosphere due to absorption of longwave radiation by the volcanic aerosols is simulated, although the temperature anomalies (Vol1 - Ref1) up to 16 K are comparatively high. The Pinatubo eruption produced a much weaker signal, but we have intentionally chosen such a strong forcing to identify potential dynamical mechanisms qualitatively. The model simulates a warming of the polar summer mesopause region of up to 15 K that can be explained plausibly with atmospheric dynamical coupling processes. The peak of the mesopause warming is reached 5 to 6 months after the eruption, which agrees with some observations that reported an equally fast response after the 1991 Pinatubo eruption. We argue that the temperature gradient in the stratospheric summer hemisphere changes the filtering of upward propagating gravity waves. This reduces the net deposition of westerly momentum by gravity waves in the upper mesosphere and thus the upwelling of air over the summer pole leading to a positive temperature anomaly. Intrahemispheric coupling (confined to the summer hemisphere) is the dominant dynamical mechanism. A second experiment (Ref2 and Vol2) with an altered parameterization of sub-grid scale orographic effects was performed to test the sensitivity of the mesospheric perturbation to differences in the unperturbed climatological state of the circulation. A comparison between both experiments hints at a possible contribution from interhemispheric coupling. This experiment simulates a temperature anomaly of 21 K for the summer mesopause, i.e. a qualitatively similar response of the summer mesopause as the first experiment (Ref1 and Vol1), despite the differences in the winter stratospheric response. We therefore conclude that interhemispheric coupling has a relativley small contribution while intrahemispheric coupling seems to be the dominant mechanism in our simulation.

Volcanic forcing is an important contribution to summer mesopause variability. Other factors such as the quasi 2-day-wave (Offermann et al., 2011; Craig et al., 1980; Pendlebury, 2012), the quasi 5-day wave (e.g., von Savigny et al. (2007)), solar proton events (Becker and von Savigny, 2010) and effects of sudden stratospheric warmings on the mesopause temperatures above (Yang et al., 2022) are already discussed in the literature, but would benefit from longer observational time series and more high-top model simulations. Findings such as those presented in this study are especially relevant for the noctilucent cloud community, because these upper atmospheric clouds are sensitive to the mesopause temperature.

*Author contributions.* HS, CvS and SW designed the study, SW prepared the initial draft of the manuscript and all authors edited and revised the manuscript.

*Competing interests.* The authors declare no competing interests

330 *Acknowledgements.* This study is part of the VolDyn project that is embedded in the research unit VolImpact (grant no. 398006378) and funded by the German Research Foundation (DFG). SW also wants to thank Ulrike Niemeier, Henning Franke, Felix Bunzel, Andrea Schneidereit and Clarissa Kroll from the Max Planck Institute for Meteorology in Hamburg for valuable discussions on the TEM diagnostics. The TEM diagnostic was done using a script that was written by Felix Bunzel and altered by Henning Franke. This work used resources of the Deutsches Klimarechenzentrum (DKRZ) granted by its Scientific Steering Committee (WLA) under project ID bb1093.

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
