# Peer review of "Impact of a strong volcanic eruption on the summer middle atmosphere in UA-ICON simulations"

_EGUsphere, 2023_

## Author Comment (AC1)

**Answers to the Reviewers**

**Review from Anonymous Referee 1:**

Review of "Impact of a strong volcanic eruption on the summer middle atmosphere in UA-ICON simulations" by Wallis et al. The manuscript is focused on understanding the response of the summer mesosphere to a strong volcanic eruption. These effects are investigated using the UA-ICON model, with the volcanic effects included by simulating the influence of an injection of 20 Tg S into the stratosphere. The simulations indicate that a large response ($\approx$15-20 K) occurs in the summer mesosphere several months after the simulated eruption. Two sets of ensemble simulations with different gravity wave forcing are used to diagnose the mechanism by which the volcanic eruption influences the summer mesosphere, with a particular focus on inter- versus intra-hemisphere coupling. The manuscript provides insight into how the mesosphere responds to volcanic eruptions, and would be suitable for publication. However, I believe that there are a number of aspects that would first need to be addressed prior to publication. These are provided in the specific comments below.

The authors thank the Anonymous Referee for taking the time to review our manuscript. We find the comments very helpful and will address them below.

Major Comments

1. The manuscript would benefit from additional description of how the volcanic eruption is simulated in the model. Although a description is provided in Section 2.2, the reviewer found it difficult to understand exactly how the effects of the volcanic eruption are included. My interpretation from the text is that this is done by specifying a modification of the aerosols in the model, which then influence the stratosphere heating. It is recommended that the authors revise the description of the simulation setup in order to make the description of how the volcanic eruption is included in the model clear to the reader. It would also be beneficial to explicitly state the timing of the simulated eruption, which can only be inferred from the text and figures currently.

We agree that the description of the volcanic forcing used in the simulation needs to be expanded. We used the EVA module to generate aerosol optical properties (such as wavelength-dependent aerosol extinction) that are equivalent to an idealized tropical eruption that emits 20 Tg $SO_2$ into the lower stratosphere. We rewrote the paragraph: "A volcanic forcing representative of a tropical volcanic injection of 20 Tg S into the lower stratosphere was generated offline by the Easy Volcanic Aerosol (EVA) forcing generator which is described in detail by Toohey et al. (2016). This relatively large amount of approximately twice of what was emitted in the 1991 Pinatubo eruption (Guo et al., 2004) was chosen to obtain a large signal-to-noise ratio. The EVA module determines optical properties (aerosol extinction, asymmetry parameter and single scattering albedo) of an idealized stratospheric volcanic aerosol distribution which itself is calculated in a simple three-box model using eruption location, and date, and amount of injected sulfur as input parameters. The model was originally tuned to represent the distribution resulting from the Pinatubo eruption."

2. There are clear differences in the results for the two experiments with different gravity wave parameters. However, it is unclear how to interpret these results. My understanding is that the results in experiment 1 use the default gravity wave parameters, which were presumably tuned to obtain accurate model climatology, but that using modified gravity wave parameters provides responses to the volcanic eruption that are more consistent with expectations, especially in the response of the Northern Hemisphere polar vortex. The results would thus partly seem in conflict. That is, the tuned gravity wave parameters would give a better climatology but potentially worse volcanic eruption response, while the modified parameters give worse climatology but better response to the eruption. Is

this correct? It is recommended that the authors include some additional discussion with regards to how to interpret the results with the two different specifications of the gravity wave parameters.

The tuning of experiment 1 was chosen as in Borchert et al. (2019) and Giorgetta et al. (2018) to ensure comparability. Their tuning choices were made for an overall acceptable model performance. Parameters of experiment 2 were tuned with the specific intention to simulate a stronger, more realistic NH polar vortex. This tuning also causes the model to simulate a more typical response to volcanic forcing which is characterized by a vortex strengthening. However, for our purpose it is very useful to have two model configurations which simulate very different winter vortex responses because this allows us to conclude on the potential influence of these different stratospheric responses on the mesospheric response. We have modified the manuscript both in the experiment description and the presentation of the results to make the motivation for performing experiments with different tuning and the conclusions they allow for the contrinution of hemispheric coupling more understandable.

3. The UA-ICON model does not include interactive chemistry. This represents a possible limitation to the simulations. For example, the effects of the volcanic eruption on ozone are not included. This limitation is not discussed at all in terms of how to interpret the results. Additional discussion of the potential limitations of the study due to neglecting the chemical effects should be included.

We agree that the limitation of UA-ICON due to its lack of interactive chemistry should be stated and discussed. We included an appropriate paragraph: "The UA-ICON model does not calculate atmospheric chemistry interactively, e.g. it does not account for heterogenous chemical reactions on volcanic sulfate aerosols. In the atmosphere, those reactions could deplete nitrogen dioxide (Aquila et al., 2013), activate chlorine (Solomon et al., 1999) and hence change the ozone concentration in the stratosphere (Rozanov et al., 2002). Ozone is important for the radiation balance in the atmosphere and changes in its concentration could affect the temperature and subsequently the atmospheric circulation. Rozanov et al. (2002) used the University of Illinois at Urbana-Champaign (UIUC) stratosphere–troposphere general circulation model to separate the volcanic impact of radiative aerosol heating and ozone depletion due to heterogenous chemical reaction on sulfate aerosols. They found that the temperature anomaly in the lower stratosphere is predominantly caused by the radiative heating of the aerosols and not the relatively small cooling by volcanic ozone depletion. These results are supported by Kilian et al. (2020) in their chemistry-climate model EMAC. The warming of the tropical lower stratosphere itself will probably alter the transport of ozone and in turn have an impact on the temperature state of the atmosphere. UA-ICON is run with a prescribed climatological field for the concentration of ozone. Hence, the omission of sulfate aerosol interaction with ozone and also the neglect of chemical transport as a result of changed atmospheric circulations are a clear limitation of this model. We would argue, however, that the qualitative argument of our study is still valuable to explain the dynamic response of the atmosphere due to volcanic eruption."

4. The interpretation of the results in terms of the effects of a large volcanic eruption on the mesosphere are unclear. Should the effects in terms of the summer mesosphere cooling be considered only qualitatively? That is, the results of the study show the potential mechanisms that would lead to the summer mesosphere cooling, but the magnitude of the cooling is uncertain.

There are only few observations of the mesosphere after strong, explosive eruptions, mainly the 1991 Pinatubo eruption. These were sensing from the tropics to the midlatitudes and some of them indicate a warming of the upper mesosphere. We are therefore, unfortunately, not able to validate the results of our study to observations of the polar summer mesopause so far. Moreover, we chose a very strong forcing (equivalent to approximately twice of the $SO_2$ emission as the Pinatubo eruption) and

simulated a temperature anomaly in the lower stratosphere that seemed particularly high. We would like to treat our simulated temperature anomaly as a qualitative result. We think there is scientific value in the identification of the dynamical processes and consider this one of the main results from our study.

Minor Comments:

1. Line 50: "below as as" should be "below as"

Thank you, this typo is now fixed.

2. Line 66: "the dynamic core" should be "the dynamical core"

We changed the text accordingly.

3. Line 86: The authors should clarify that the two reference experiments are also ensemble simulations.

We edited the sentence as follows: "We additionally performed non-volcanic ensemble simulations as reference for each of the two experiments (Ref1 and Ref2)."

4. Lines 111-112: The authors should consider moving this text to the beginning of Section 3 so that it is immediately clear to the reader why the results in Figure 3 are focused on November-February.

We moved this text to the beginning of the result section.

---

## Author Comment (AC2)

**Answers to the Reviewers**

**Review from Anonymous Referee 2**

Impact of a strong volcanic eruption on the summer middle atmosphere in UA-ICON simulations

Sandra Wallis, Hauke Schmidt, and Christian von Savigny

This paper focuses on the response of the mesosphere to heating in the lower tropical stratosphere that is caused by an injection of aerosols from a large volcanic eruption. To do this a high-top GCM is used where the volcanic forcing is prescribed. Analysis of the differences in ensemble means of runs between those with and without volcanic forcing indicates the mesosphere in the Southern Hemisphere summer changes. This is attributed to changes in momentum deposition from parameterized non-orographic gravity waves. The sensitivity to the response to changes in sub-scale orographic drag parameterization settings is tested. The paper is well written and the presentation of the results is, for the most part, clear. Its topic is suitable for ACP. However, the methodology needs to be better explained and I have some concerns about interpretation. Specifically, the paper could be improved in several areas by providing: 1) a better model description (esp. the handling of radiative transfer), 2) analysis of the mesopause temperature and location, rather than temperatures in the mesopause region, 3) analysis of the drivers of stratospheric wind changes (e.g., EP flux divergence vs thermal wind changes), 4) an explanation of the orographic gravity wave parameterization and motivation for changing it (with respect to inter vs into hemispheric coupling), and 5) an expanded description of the model volcanic forcing. Below are specific comments that I think if addressed will cover most of the issues listed above.

*We would like to thank the referee for taking the time to review our manuscript. We find the comments very helpful and will address them below.*

Specific comments:

The Abstract needs updating to reflect the results of the study in a quantitative sense. Almost all the text is devoted to motivation and saying what will be done ("This study will focus"), rather than what was done. There is no mention of the sensitivity of test with the orographic drag parameterization.

*As the reviewer suggested, we included a reference to the sensitivity experiments ("Two experiments with differently parameterized effects of sub-grid scale orography are compared to test the impact of different atmospheric background states.") and changed the tense of the last sentence ("This study focuses" instead of "This study will focus").*

L8: "The simulation" does not reflect that ensembles were run with different gravity wave parameterization settings.

*To make it clear to the reader that we refer to two experiments (with different gravity wave parameterization) we changed "The simulation" to "The simulations" referring to both experiments mentioned in the sentence before (see reply to the comment above).*

L63-75: The description of the UA-ICON is insufficient to determine the suitability of the model for the current study. Many details are missing. For example, what was changed in the dynamical core to go "from shallow to deep atmosphere dynamics" (L67). For what date was IFS analysis used to initialize the model and how was the model initialized all the way up to 150 km? How are the prescribed constituents used (L71)? How is radiative transfer handled, are non-LTE effects and chemical heating accounted for? What is the source of the solar spectral irradiance and what wavelength region does it cover? If this is the first time this model has been presented, some validation should be shown

(e.g., comparison to URAP or another satellite climatology).

We added more details on the UA-ICON model in our "tools and experiments" section: "The shallow approximation neglects terms related to the spherical curvature of the atmosphere as well as the variation of the gravitational field. These terms are modified in order to account for a more realistic deep atmosphere (Borchert et al., 2019). Furthermore, a physics package for the upper atmosphere is implemented. This study uses a R02B04 grid, i.e., a triangular grid with about 160 km horizontal mesh size, with 120 vertical levels and a model top at 150 km altitude. Each run is initialized from an Integrated Forecasting System (IFS) analysis provided by the European Centre for Medium Range Weather Forecast (ECMWF). Our first ensemble member uses the IFS products starting in the year 1980, the second ensemble member started in 1981 etc. The IFS model has a model top at approximately 0.01 hPa and is used to initialize the state of the lower and middle atmosphere in UA-ICON. Above, vertical temperature profiles from Bates and Massey (1959), Hedin (1983) and Fleming et al. (1988) are used as initialization for the upper atmosphere and to calculate pressure and horizontal winds for that region. The UA-ICON model does not simulate atmospheric chemical reactions, but prescribes climatological fields for concentrations of $O_3$, $O_2$, $O$, $NO$, $CO_2$, $CH_4$ and $N_2O$. Additionally, climatologies are prescribed for tropospheric aerosol optical properties, the sea surface temperature and sea ice. The air masses in the upper atmosphere are no longer in local thermodynamic equilibrium and modifications included in the physics package are necessary. Non-local thermodynamic equilibrium cooling of $CO_2$ and ozone are implemented above 65 km using Fomichev and Blanchet (1995) (with modification from Fomichev et al. (1998)), the $CO_2$ absorption above 19.25 km using Ogibalov and Fomichev (2003) and the NO cooling above 60 km by Kockarts (1980). A 35-year HAMMONIA simulation provides the monthly and zonal mean chemical heating rates that are used in UA-ICON. Spectral solar irradiance is prescribed at the model top. UA-ICON uses the PSrad radiation package (Pincus and Stevens, 2013) (covering wavelengths longer than 200 nm) as the standard ICON model, but additionally implements the UA package. Above 50 km, this package uses a model by Strobel (1978) to account for the $O_2$ Schumann-Runge bands from 175 to 205 nm and the continuum from 125 to 175 nm. Above 90 km, a model based on Richards et al. (1994) is used to include the extreme UV range from 5 to 105 nm. A validation of UA-ICON against the Sounding of the Atmosphere using Broadband Emission Radiometry (SABER) on the Thermosphere Ionosphere Mesosphere Energetics Dynamics (TIMED) satellite and against a climatology from Upper Atmosphere Research Satellite Reference Atmosphere Project (URAP) is given in Borchert et al. (2019)."

L74: described -> parameterized or implemented?

We replaced "described" with "implement"

L76: A more detailed description of the 'volcanic forcing' is needed. It was not clear if the aerosols distribution is prescribed or prognostic. Does it react at all to the change in tropical upwelling rates shown Figure 3? How does the vertical distribution evolve with time? More importantly, is it realistic? Unless I missed it, the date of the eruption is not described, nor how long the model ran before the eruption.

We used the Easy Volcanic Aerosol (EVA) module by Toohey et al. (2016) to prescribe the radiative forcing from the sulfate aerosols. The EVA module uses the eruption location, the date and the estimated stratospheric sulfur injection as an input to calculate stratospheric aerosol optical properties offline. These properties are the aerosol extinction, the single scattering albedo and the asymmetry parameter. The aerosol extinction allows to determine the aerosol optical depth that is important for

the radiative transfer. The EVA module only provides idealized volcanic forcing. A comparison for a EVA generated Pinatubo forcing with the Chemistry-Climate Model Initiative forcing set (using SAGE II extinction) is shown in Toohey et al. (2016). The vertical distribution does evolve in time but it does not interact with the tropical upwelling rates in the UA-ICON model runs. We edited the text accordingly: "A volcanic forcing representative of a tropical volcanic injection of 20 Tg S into the lower stratosphere was generated offline by the Easy Volcanic Aerosol (EVA) module (Toohey et al., 2016)." and "The volcanic runs in Vol1 and Vol2 were all started in May and the eruption was simulated on June 15 as a reference to the Pinatubo eruption."

L87-91: I found the sensitivity experiment (settings in the Lott parameterization) poorly motivated? Why is it necessary to explore the impact of "relatively different representations of the polar vortex". The reference to Figures 4 and 8 do not actually show the state of the polar vortex, so it is difficult to judge what or why this matters. Saying the G and Cd changes also is not helpful to a reader who is familiar with the parameterization. Please describe what these parameters do and describe how the model state has changed rather than asking the reader to try to determine the differences between two figures. For example, I would guess that Cd is a (surface?) drag coefficient? Is it reasonable to change it by a factor of 2? How long was the model allowed to adjust to the new settings from its IFS initialization?

The ensemble mean of the first experiment (Ref1 and Vol1) does not show a strengthening of the NH polar vortex as we expected based on the experience from simulations with other models and occasional observational evidence. A comparison between Figure 4 and 8 rather hints at a weakening of the vortex. The sensitivity test was done by altering the sub grid scale parameterization in a way to generate a circulation with an on average stronger polar vortex. Indeed, this stronger background vortex leads also to the expected response to volcanic aerosol of a further vortex strengthening. To understand the causes of this dependence of the vortex response on the background state could be of interest in a study focusing on the stratosphere, but is not relevant for our mesospheric focus. However, the fact that the summer mesospheric response is similar despite the differences of the response in the winter stratosphere allows us to conclude that interhemispheric coupling is not the dominant dynamical mechanism in our study. Since the response of the winter stratosphere is different for both experiments, but the qualitative response in the polar summer mesopause is comparable, we concluded that interhemispheric coupling is not the dominant dynamical mechanism in our study. We also edited the paragraph describing the sub-grid scale parameterization to improve its clarity: "We used two different settings of the Lott (1999) parameterization of sub-grid scale orographic effects chosen in order to have experiments with different representations of the polar vortex. The parameterization represents two different effects of unresolved orography: a) low-level blocking of near-surface flow that would be forced to flow around the orographic barrier. The strength of his effect can be scaled by the tuning parameter $C_d$. b) Momentum transfer by gravity waves caused by the orography (scaled by parameter G). The first experiment follows the setting of Borchert et al. (2019) with G=0.1 and $C_d$=0.01, while in the second experiment, G=0.05 and $C_d$=0.2 are used. This means that the effects of atmospheric drag from gravity waves expected from unresolved orography is reduced and additionally the low-level blocking effect of unresolved orography is enhanced. Therefore, the second experiment explicitly reduces the contribution of orographic gravity wave generation." The model itself was not allowed to adjust to the new settings.

L93: It would be good to provide a reference for formulation of the TEM calculation used (e.g., Gerber and Manzini [2016] or Andrews, Holton and Leovy, etc.)?

We agree and added a reference to Hardiman, S. C., Andrews, D. G., White, A. A., Butchart, N.,

& Edmond, I. (2010). Using Different Formulations of the Transformed Eulerian Mean Equations and Eliassen–Palm Diagnostics in General Circulation Models, Journal of the Atmospheric Sciences, 67(6).

L99: It would be good to state the differences from the Ref1 case as well as the absolute temperature. It's not clear if this is simply an upward shift in the location of the mesopause (as indicated by dipole patter) or a cooling of the mesopause.

We would like to refer to our answer below concerning the comment on L140.

L103: Perhaps showing the w* anomaly would be more convincing, rather than streamfunction. Note, also, that Figure 3 has no units.

The units for the streamfunction in Figure 3 is indicated as a text insert at the bottom right of every graph (e.g. "(Ref1) in kt/s"). However, we admit that the font is small and this information can be easily overlooked. Therefore, we now additionally include this information in the caption: "Similar to Figure 2, but for the zonal, monthly and ensemble mean residual mass-streamfunction in units of kt/s." Below you find a figure similar to Figure 3 in the manuscript except that the residual mean w velocity is shown instead of the residual streamfunction. We argue that the strengthening of the Brewer-Dobson circulation is easier seen when the residual streamfunction is shown and would like to keep the original figure in the manuscript.

[Figure]

L107: Be sure to be consistent in the font used for v*. Also, if it is the residual velocity, I think it should have an overbar.

We agree and changed the symbols accordingly.

L119: Is the change in the zonal wind in the stratosphere only from the change in temperature gradient? I would think any change in the stratosphere would impact the wave driving. The EP flux divergence in Figure 7b seems to be non-negative but the scale does not really reveal the magnitude of the forcing.

[Figure]

A zoom-in on the SH stratosphere reveals that there is a negative EP flux divergence anomaly in the upper stratosphere with values of about - 1 m/s/d. The lower stratosphere shows a positive EP flux divergence anomaly smaller than 1 m/s/d. We agree that processes other than the zonal wind change due to the temperature gradient between the tropics and the high latitude can impact the wave driving. Nevertheless, it seems plausible that the zonal wind anomaly shown in Figure 4 would alter the gravity wave filtering. Although the complexity of the atmospheric circulation does not allow us to exclude other processes in the UA-ICON model, we think that the dynamical process discussed here is of value to explain the impact of the volcanic eruption on the summer polar mesopause.

L128: Since you are talking about non-zero phase speeds, it would be good to be explicit that you are referring to the non-orographic gravity wave forcing. Regarding Figure 5 and the 'zonal wind tendency', again, it's not clear if this is only from the non-orographic parameterized gravity waves or all Gws.

The zonal wind tendency in Figure 5 is the non-orographic gravity wave drag in zonal direction. We added this information to the text ("This is suggested by the negative anomaly of the zonal wind tendency (Vol1 - Ref1), i.e. the non-orographic gravity wave drag in zonal direction.") and the caption of Figure 5.

L140: Picking a fixed altitude in Figure 6 does not show the mesopause change, which is the cold point. What should be shown is the mesopause temperatures for each case and their differences. It would also be useful to show the height variation.

Indeed, the altitude of the polar summer mesopause for the ensemble mean is elevated up to 4 km in the first post-volcanic November, however, the variation among the ensemble members is large and include ensemble members with a descending cold point. Considering only the temperature anomaly of the mesospheric cold point, a temperature anomaly of up to 11 K is reached the first

post-volcanic winter having a maximum in February. We decided to keep the original Figure 6 in the manuscript, because it allows us to approximate the time delay between the simulated eruption and the maximum temperature response in the mesopause region. In our opinion, this approach of showing the temperature anomaly can be compared more easily with (midlatitude) observations, which will be discusses further (and with more examples) in the discussion section. Nevertheless, we included the figure below in the SI.

[Figure]

Figure S1: Time series of the (a) altitude and (b) temperature anomaly of the mesospheric cold point. Gray shading indicates 1 standard deviation of the 10 ensemble member anomalies (Vol1 - Ref1). The date of the eruption as well as of the first and second post-eruption Novembers are marked with dash lines.

L153: I think it is generally acknowledged that wave-mean flow interactions drive the stratospheric circulation and gravity waves are responsible for the driving of the deep branch of the B-D circulation and the cold summer mesopause. It is therefore not surprising that the EP flux divergence is relatively small at the mesopause. What I think it missing here is a quantification of the impact of the resolved wave driving on stratospheric zonal winds that then impacts the wave filtering of the parameterized gravity wave forcing. It seems too simplistic to simply say the heating in the tropics causes a thermal wind response at mid-latitudes that impacts gravity wave filtering.

Figure 7b shows the Eliassen Palm flux divergence for November and indicates a negative anomaly in the polar vortex region, e.g. a weakening of the vortex. We are, however, aware of the interaction between waves and the mean flow and therefore found the thermal wind argument usefull in our discussion. There are some papers that discuss this subject in more details, such as Bittner et al. (2016). In contrast, our study focuses on the mesosphere.

L156: Can you clarify why "the simulated weakening response of the polar vortex in the winter stratosphere seems unrealistic"?

We added the sentence: "Additionally, the simulated weakening response of the polar vortex in the winter stratosphere seems unrealistic when it is compared to the typical response of GCMs to aerosols from strong tropical volcanic eruptions (Bittner et al., 2016)."

Bittner, M., Schmidt, H., Timmreck, C., and Sienz, F. (2016), Using a large ensemble of simulations to assess the Northern Hemisphere stratospheric dynamical response to tropical volcanic eruptions and its uncertainty, Geophys. Res. Lett., 43, 9324– 9332, doi:10.1002/2016GL070587.

L156: "the gravity wave parameterization was therefore changed" is ambiguous - please be explicit that you are changing the orographic GWs and why.

We agree and included the word "orographic" to improve the clarity of the sentence.

L160: This is confusing - the volcanic forcing causes the anomaly. Did the mesopause really warm 21K? In other words if the mesopause was 140K it only gets down to 161K?

We changed the text to be more specific: "The altered gravity wave parameterization results in a significant positive temperature anomaly of up to 21 K in the polar summer mesopause region from 80 - 95 km altitude and between 70°S - 90°S in November (Vol2 - Ref2, see Figure 9b) supporting the results of the former experiment.".

L164: It is not clear if this difference of differences is coming from the difference in the Refs (very likely) or that the *response* was modified by the choice of the orographic gravity wave forcing. Which is it?

The sub-grid scale parameterization was altered between experiment 1 (Ref1, Vol1) and experiment 2 (Ref2, Vol2). By comparing the differences of the anomalies, e.g. (Vol2 – Ref2) – (Vol1 – Ref1), we aim to show how the volcanic response differs for the different backround states simulated for the different parameter settings. Figure 10 indicates small differences in the mesopause responses for the two different experiments, but the pattern of the difference of temperature, gravity wave drag and the residual streamfunction hints at inter-hemispheric coupling as a possible cause of this small effect. We therefore find it valuable to show these comparisons and discuss its possible implications. We also agree with the referee that the choice of the parameterization is not solely affecting the volcanic response, but also the state of the atmosphere in the reference runs. The figures below show the difference between the ensemble mean reference temperatures (left) and the difference of the temperature anomalies as presented in the paper (right). It is apparent that the altered parameterization of the sub-grid-scale orography results in differences between the reference runs in both experiments. We therefore added a comment to the discussion section and added a plot with the difference of the reference in the SI: "Moreover, a change of the sub-grid scale orographic parameterization is not only affecting the volcanic perturbation, but also leads to differences in the volcanic-free reference runs (e.g. the temperature, see Figure S3 showing Ref2 - Ref1)." Nevertheless, we argue that it is of value to compare the volcanic response (Vol - Ref) of both experiments to estimate the contributions of intra- and interhemispheric coupling.

[Figure]

L212: Do you really mean blocking here? Or filtering? Blocking typically refers to a surface pressure condition.

Yes, we are referring to the concept of low-level flow blocking that is used for the sub-grid scale parameterization as described in Lott (1999). We changed "low-level blocking" to "low-level flow blocking" to be more precise and use the terminology earlier in the manuscript when the different tunings of the Lott (1999) parameterization are described.

L266: The explanation of how the Ref2 and Vol2 experiments can isolate the interhemispheric coupling needs to be expanded - can you describe how this works? Preferably much earlier in the paper. Apologies if I missed the argument. As far as I can see this just changes the strength of the zonal wind in the Southern Hemisphere.

Initially, we performed the second experiment because the first experiment did not show the expected polar vortex strengthening as response to volcanic forcing. Earlier work has shown that the winter stratosphere can influence the summer mesosphere via interhemispheric coupling, so that an unrealistic response of the winter stratosphere to volcanic forcing could also compromise the summer mesospheric response. Hence, the second experiment was designed with the intention to create a different, and hopefully more realistic, response of the winter stratosphere and thereby test the robustness of the mesospheric response under different stratospheric responses, or, more generally, under different circulation states of the reference atmosphere. We agree that the original sentence in the manuscript needs rephrasing. We changed it to: "A second experiment (Ref2 and Vol2) with an altered parameterization of sub-grid scale orographic effects was performed to test the robustness of the mesospheric response to volcanic forcing under different background states of the circulation. A comparison between both experiments hints at a possible contribution from interhemispheric coupling." This better reflects the argument given in the discussion section.